# For the Sake of the Future: Can Democratic Deliberation Help Thinking and Caring about Future Generations?

**Katariina Kulha** [1], **Mikko Leino** [1], **Maija Setälä** [1,*], **Maija Jäske** [2] and **Staffan Himmelroos** [3]

1 Department of Philosophy, Contemporary History and Political Science, University of Turku, 20500 Turku, Finland; kasuku@utu.fi (K.K.); molein@utu.fi (M.L.)
2 Social Science Research Institute, Åbo Akademi University, 20500 Turku, Finland; maija.jaske@abo.fi
3 Department of Politics and Communication, Helsinki University, 00100 Helsinki, Finland; staffan.himmelroos@helsinki.fi
* Correspondence: maiset@utu.fi

**Abstract:** This article examines whether democratic deliberation can enhance participants' capacity to consider future generations' perspectives and willingness to make sacrifices to ensure their well-being. In addition to normal deliberation, we are interested in the effects of a mental time travel exercise where deliberators imagine themselves in the future (without ageing). The study is based on an experiment conducted as a part of Citizens' Assembly that contributed to the long-term planning of the Satakunta region in Finland. Our findings suggest that deliberation as such increases participants' willingness to consider future generations' perspectives in long-term planning; yet the mental time travel exercise had only a modest impact on perspective-taking. The results also show some support for the assumption that deliberation can enhance willingness to make sacrifices for future generations, although we do not see such an impact in case of an intergenerational conflict in flood protection.

**Keywords:** deliberation; future generations; future design

## 1. Introduction

The need and urgency of sustainability transformations presents democratic societies with an unforeseen challenge. Decisions concerning climate change, biodiversity loss and other environmental problems should satisfy the needs of citizens living today and simultaneously guarantee that the future generations will also have at least a sufficient amount of resources available [1]. The transition to a sustainable society is not just a technical endeavor but entails complex normative and political choices [2]. Without citizen participation and inclusive decision-making processes, these choices carry the risk of coming out as illegitimate [3].

It is, however, widely recognized that democratic decision-making and practices of representative democracy, in particular, are prone to short-sightedness, or democratic myopia. There are 'drivers' of short-termism in representative democracies [4] such as short electoral cycles, and the influence of interest groups that shorten the time horizon of elected representatives. Individual voters are equally prone to democratic myopia, which decreases public support for future-oriented policies. Thus, the capability of representative democratic institutions to make sustainable policy is hindered by strong political and economic interests in preserving status quo [5].

Sustainability transformations seem to require wider public appreciation of the principle of intergenerational fairness. Therefore, different modes of incorporating citizens' perspectives into sustainability transformation policies should be explored and their effects examined. Well-designed processes of citizen deliberation have been regarded as a remedy for the tendencies towards short-termism in representative systems [6]. In other words, deliberative mini-publics or randomly selected second chambers could arguably be inclusive and democratic means of enhancing sustainability transformations.

This study asks whether participation in democratic deliberation can help increase citizens' sense of intergenerational fairness. More specifically, we ask whether participation in a deliberative mini-public helps imagine and consider future generations' perspectives and increase willingness to make sacrifices for future generations' well-being. We consider acknowledgment of the needs of future generations as a precondition for willingness to allocate some portion of resources to well-being of those who come after us. Previous research shows that deliberative mini-publics can both change the minds of those taking part [7–9] and increase their knowledge and willingness to contribute to collective action [10].

In this article, we explore two separate mechanisms that might take place in a deliberative process. The first pertains to the capacity of democratic deliberation to help participants consider others' perspectives, including those of future generations. The second mechanism is that democratic deliberation should give more weight to vital interests and thus increase concern for intergenerational fairness. This in turn would make participants more willing to make sacrifices in order to ensure future generations' wellbeing. In addition, we are interested in the effects of a 'mental time travel' exercise where participants are actively encouraged to take perspectives of future people in the context of a deliberative process.

The empirical analysis is based on the experimental procedure conducted in conjunction with Satakunta2050 Citizens' Assembly. This online mini-public was organized to contribute to a long-term regional planning process reaching to the year 2050 in the Satakunta region in South West Finland. In addition to the standard procedures applied in deliberative mini-publics [11], the Citizens' Assembly entailed an experimental treatment where deliberators were actively encouraged to imagine and to consider perspectives of people living in the future. In this particular case, the participants took a mental time travel exercise where they imagined themselves in the year 2050 without ageing [12].

The next section of this article provides an account of previous theoretical and empirical research regarding the capacity of democratic deliberation to help take future generations' perspectives and consider their interests, and formulates four hypotheses on this basis. This is followed by a description of the Satakunta 2050 Citizens' Assembly, as well as the experimental procedure and its results. Our results lend some support for the hypotheses that deliberation increases the willingness to consider future generations' perspectives and make sacrifices for future generations. These results are discussed in the final section of this article.

## 2. Theory: Why Deliberation Might Enhance Thinking and Caring about Future Generations

Democratic deliberation can be understood as an intersubjective process of mutual justification among a diverse group of citizens or their representatives. Previous literature has articulated the argument that democratic deliberation can enhance foresight and consideration of long-term consequences of public decision-making [6] (p. 287). First, deliberative processes entail consideration of relevant *factual* information on policy consequences. This is likely to enhance increased awareness of the long-term effects of different policies, including the effects on future generations [4]. In this respect, what matters is the capacity of deliberative processes to facilitate learning and understanding of relevant information, especially on causal effects of different policy choices. This should help deliberators consider the possible positive or negative consequences that public policies might bring about in the future.

Second, deliberative processes are based on weighing of *normative* arguments by their merits. Self-serving arguments appealing merely to narrow, short-term interests should not be influential in a deliberative process [6,13]. This should benefit more collectivist and sustainable orientations toward public policy. Deliberative process should also help make intersubjective evaluations of the urgency of claims made [14] Arguments appealing to vital interests of individuals and groups or to major injustices between different groups should carry weight in processes of democratic deliberation. Generalizable arguments, including arguments referring to intergenerational justice, are likely to be brought up and

sustained in the deliberative process. In this way, appeals to interests of future generations, should emerge and succeed in the deliberative process.

In order to make judgments on the weight of different normative claims, deliberative processes require a capacity to understand others' perspectives, or 'to put oneself to others' shoes' [15]. Empirical studies on deliberative mini-publics show that deliberation can also enhance mutual understanding and perspective-taking among people who do not initially share similar views or identities, thus counteracting empathy biases [16,17]. Inclusive deliberation could therefore be expected to enhance empathy towards 'out-groups' represented in the deliberative process, especially if they appear to be in a disadvantaged position. There are also some indications that deliberation can enhance consideration of perspectives of such people who are not involved in the process. A study analyzing perspective-taking (or cognitive empathy) in the context of a citizen deliberation experiment on immigration [18] concludes that participants became more willing to consider immigrants' views, even when no immigrants were present in the group. In other words, supporters of immigration seemed to have acted as 'representatives' by bringing up immigrants' viewpoints.

However, Lindell et al. [19], who study the same deliberative experiment, find out that the actual presence of an immigrant in a deliberative small group contributes to the attitude shifts to more pro-immigration direction. The finding by Lindell et al. [19] seems to highlight the importance of the 'politics of presence' in Phillips' [20] terms. Physical presence in deliberative processes seems to be important for empathic concern and perspective-taking, especially when it comes to of individuals representing different social groups [21]. Ackerman and Fishkin [13] (p. 141) argue about the importance of physical presence in deliberative forums as follows: "[B]eing in a room with randomly assigned fellow citizens can stimulate understanding across social cleavages".

Compared to other affected groups who are not represented in the deliberative process, the difficulties of taking the perspectives of future generations seem to be even more severe. This is because future generations are not just present, but actually non-existent [20]. In Phillips' terms, future generations and their interests can only be represented in the deliberative process in the sense of 'politics of ideas'. It may be hard to consider future generations' perspectives because they are necessarily hypothetical or imagined. Moreover, the scope of empathetic reactions is particularly limited because of future people do not have identities [22]. For example, artistic ways of articulating future generations' perspectives could, however, facilitate internal processes of imagination and reflection or, in Goodin's [23] terms, 'democratic deliberation within'.

Despite the reservations put forward above, we set the following hypotheses regarding the effects of deliberation on participants' capacity to take future generations' perspectives and sympathize with them.

**Hypothesis 1.** *Participation in a deliberative process helps consider future generations' perspectives.*

There is some evidence that processes of group deliberation enhance other-regarding attitudes, even individual belief in and personal proneness to collective action for goals accepted by all. Setälä et al. [10] show that participation in a deliberative mini-public can increase participants' belief in collective action to resolve societal problems and their willingness to participate in such collective action. In a more recent study, MacKenzie and Caluwaerts [24] find that deliberative practices can be used to encourage climate action policies, which are highly relevant to the interests of future generations. Based on a large-scale deliberation experiment they show that deliberation make participants more supportive of policies that help mitigate climate change and more willing to make personal sacrifices, such as paying higher gas taxes. In this this respect, group deliberative processes seem to be different from individual deliberation, which, according to studies (e.g., [25]), can even breed callousness.

While there seem to be reasons to believe in the capacity of democratic deliberation to help consideration of future policy consequences and future generations' interests, there

may be limits to this capacity. The problem of short-termism is likely to be prominent in policies that involve intertemporal welfare trade-offs, i.e., the sacrifices of welfare at the present in order to gain benefits in the future. There are different reasons for this. First, uncertainty regarding future consequences of policies is likely to give more weight to current benefits and losses in comparison to those materializing in the future [4]. Second, the abstract nature of future events, especially when expected to occur in distant future, reduces individuals' willingness to take them into proper consideration [26].

Third, deliberation may fail to give equal consideration of future generations' interests in case of so-called intergenerational conflicts. The problem of intertemporal trade-offs is likely to be especially pertinent in situations where current sacrifices are expected to be made, not in order to ensure the welfare of future selves but rather that of future others. Therefore, intergenerational conflicts seem to pose a particular challenge to the idea of intergenerational fairness. Policy proposals that entail welfare tradeoffs between current and future generations are particularly difficult to handle equitably, even in a good-quality deliberative process. There are no guarantees that future generations' interests are articulated or gain the attention they deserve in a deliberative process, not least because the bearers of those interests are not there to defend their views.

Despite these reservations, drawing on the assumptions of deliberative theory together with findings by Setälä et al. [4] and MacKenzie and Caluwaerts [24], we form the following hypothesis regarding the effects of deliberation to participants' willingness to take future generations' interests into account and even make sacrifices to ensure those interests.

**Hypothesis 2.** *Participation in a deliberative process increases willingness to make sacrifices for future generations' interests, even in intergenerational conflicts.*

*Facilitating Intergenerational Fairness through a Mental Time Travel Exercise*

Some earlier studies suggest that imagining oneself in the role of future generations as a part of deliberation might have an effect on individuals' policy preferences and other outcomes of deliberation. Visualizing the future through 'time travelling' exercises is often used in context of different future exploration activities, such as scenario building, backcasting and forecasting [27–30]). The aim of these exercises is to help participants create a picture or pictures of the future with their imagination [29].

Hara et al. [12] used a guided time travel exercise—a so-called future design—in deliberative workshops where participants' task was to create a future vision for their hometown Yahaba in Japan. They found out that groups who did the time travel exercise and assumed the position of future generations set markedly different policy goals for the development of their hometown compared to groups who did not 'time travel'. In another study, Uwasu et al. [30] obtained similar results with a corresponding future design setting.

However, there is still little research on whether such exercises, used in the context of democratic deliberation, could have an impact on individuals' capacity to take future generations' perspectives and contribute to their welfare. To explore this question, a special visualization exercise was designed for the purposes of the study. In this mental time travel exercise, participants were instructed to imagine themselves in the future, without ageing. Because such exercise should help deliberators in role-taking, or put themselves in the shoes of those living in the future, we can assume that a mental time travel exercise can enhance consideration of future generations' perspectives and increase willingness to contribute to their welfare, even in intergenerational conflicts.

Based on these earlier studies, the following two hypotheses are formulated:

**Hypothesis 3.** *Mental time travel in conjunction with a deliberative process helps participants consider future generations' perspectives.*

**Hypothesis 4.** *Mental time travel in conjunction with a deliberative process increases participants' willingness to make sacrifices for future generations' interests, even in intergenerational conflicts.*

### 3. The Experimental Design: The Satakunta2050 Citizens' Assembly

*3.1. The Purpose of the Citizens' Assembly and the Recruitment of Participants*

The data utilized in this study originates from a Regional Citizens' Assembly organized online in the fall of 2020. The Citizens' Assembly was organized by researchers in collaboration with the regional authorities. The assembly took place in Satakunta, a region with a population of around 224,000, located in South West Finland. The Citizens' Assembly process was linked to the drafting of Regional Strategy for 2050, a work carried out by the Regional Council of Satakunta. Regional Strategy is a document prepared according to the Land Use and Building Act (132/1999) [31] in Finland, outlining the general development trajectories and goals of the region for the coming decades. Regional Strategy is an integral part of steering official activities towards sustainability. As highlighted in the Act on Regional Development and Administration of Structural Funds (7/2014, 4§) [32], sustainability should be one of the core features in the economic and environmental development of the regions.

The purpose of the Citizens' Assembly was to allow citizens living in the region to take part in the drafting process of the Regional Strategy. In what follows, we first outline the recruitment process and composition of the Citizens' Assembly, and explain the experimental design thereafter. The recruitment of participants began in February 2020 with a recruitment survey mailed to a random sample of 6000 residents aged between 15 and 80 living in Satakunta region. The survey included questions related to the preferred future of the region, measures on respondents' political future orientation and tendency to think about future generations' interests, in addition to enquiries about socio-economic attributes. Respondents also had the opportunity to indicate whether they would be willing to participate in the Regional Citizens' Assembly to be held in mid-April. The respondents were also informed about the compensation of 75 € for participants of the assembly. Of the initial sample of 6000, around 17 percent ($n = 1049$) responded to the survey and of the respondents, 281 volunteered to take part in the deliberative event.

However, due to the outbreak of the Covid-19 pandemic, considerable changes to the process had to be made. Following the governmental response to the pandemic, all volunteers were informed that the Citizens' Assembly would be postponed until fall of 2020. In June, the researchers decided to move the assembly online. All volunteers were informed that the assembly would be carried out online using Zoom application in September, and were invited to take part in it. From this group of 281 volunteers, 70 people expressed their willingness to take part in the Citizens' Assembly. This is a notable decrease, which can most likely be explained by changes made in the organisation of the event. Some confirmed participants dropped out at the last moment, and 55 participants participated in the discussions. Basic information regarding the demographic composition of the assembly is presented in Table 1.

As can be seen from Table 1, the Citizens' Assembly represented the population of Satakunta quite well in terms of gender, age, native language and place of living. Even though there were more men than women attending the assembly, the gender balance is quite satisfactory. When it comes to age, the recruitment process managed to gather a well-rounded group, where the youngest participant was 17 years old and the oldest 79 years old. In terms of place of living, the divisions between the three sub-regional units in Satakunta is almost identical in the assembly and in real life. When it comes to education level, it must be noted that the education level among participants was higher than among the population as a whole. Especially people holding a university degree were overrepresented in the assembly.

**Table 1.** Demographics (%).

| | | Satakunta 2018 | Participants (*n* = 55) |
|---|---|---|---|
| Gender | Male | 49.7 | 56.4 |
| | Female | 50.3 | 43.6 |
| Age | Mean | 45.7 | 49.2 |
| Native language | Finnish | 96.1 | 98.2 |
| | Other | 3.9 | 1.8 |
| Sub-region | Northern Satakunta | 8.5 | 9.1 |
| | City of Pori sub-region | 60.6 | 50.0 |
| | City of Rauma sub-region | 30.9 | 30.9 |
| Education level | Mean (1–5) | 2.1 | 3.3 |

Education level coded as follows: 1 = Elementary school or middle school, 2 = Matriculation exam or vocational degree, 3 = College level vocational degree, 4 = Polytechnic degree or lower university degree, 5 = Upper university degree. (Demographics represent the situation on December 2018, from where we have complete data regarding regional demographics (see [33,34])).

The main task of the Citizens' Assembly was to prioritize different goals related to regional development, and give comments on specific future visions drafted by the Regional Council. Goals covered a broad range of policy areas: health and social services, inequality, economy, education, technical development, creating jobs, biodiversity, climate change mitigation, transport system, equality between regions and cultural heritage. After the deliberative process, participants prioritized these goals individually by selecting three that were most important to them. Here economic development and jobs, education possibilities, and a well-functioning transport system were the three most frequently selected goals in participants' prioritizations. When comparing participants' post-deliberation prioritizations to those before deliberation, the most notable change was an increased weight given to education opportunities. Participants' views and written feedback were handed to the Regional Council to be taken into account in the drafting of the Regional Strategy.

Based on participants' evaluations, the deliberative quality of the Citizens' Assembly was high, despite the move to the online environment. A vast majority of participants agreed with positive statements of the process such as 'I had enough opportunities to express my opinions during the deliberation' (92.5% completely and 7.5% partly agreed), and 'Other participants respected my views regardless of whether or not they agreed with me' (90.6% completely and 7.5% partly agreed, 1.9% could not say). In addition, 81.1% of participants completely and 11.3% partly disagreed with the statement 'I had difficulties of expressing my opinions in the discussion if others disagreed with me (5.7% agreed partly, 1.9% could not say).

### 3.2. Experimental Procedure and Data

The aim of this article is to study whether deliberation and a specifically designed 'time travel' experiment helped participants' take future generations' perspectives and become ready to make sacrifices for future generations' well-being. For this purpose, a 'time travel' exercise, or using the terminology by Hara et al. [12], future design, was planned to be carried out in conjunction with the Citizens' Assembly.

The participants were randomly allocated into a treatment condition and a control condition. In the treatment condition, participants carried out the future design exercise before deliberation, while in the control condition participants engaged in 'normal' deliberation. Deliberation took place in altogether randomly allocated 10 small groups, five in treatment and five in control condition. Each small group had an average of five or six participants. The deliberative process lasted for four hours, with a 45 min break in the middle. Apart from the 'time travel' exercise, similar deliberative procedures were followed in the small groups with future design and in groups that engaged in 'normal' deliberation. In the future design treatment, participants were also encouraged to think about the viewpoint of the people living in 2050 during the small group deliberation.

The future design exercise was composed by the researchers, following loosely the examples of Markley [27], Cuhls [29], Hara et al. [12] and Nakagawa et al. [35]. The exercise's aim was to help participants assume the perspective of future generations during the deliberation. The future design consisted of a visualization part and a short reflection phase. During the visualization, participants were instructed to 'time travel' to the future without aging. They were asked to imagine themselves in their home surroundings in Satakunta region in the year 2050. To guide the visualization process, participants were asked different questions, such as 'Do you see something familiar or something completely new?', 'Do you see other people?', 'What topics would you discuss with your close ones in 2050?', 'What would you read from the news in 2050?'. In practice, the visualization was conducted by playing a pre-recorded audio tape to the participants.

In the reflection phase, participants were instructed to write down thoughts and emotions experienced during the visualization. These reflections were not discussed jointly with the group. As an instruction for the upcoming discussion, participants were prompted to consider how today's decisions will affect people living in the year 2050. More specifically, they were asked 'What kind of actions and decisions would the people of 2050 want people living in 2020 to take and make?' The exercise was recorded in advance and played to participants by the moderator to ensure the treatment would be similar to all groups. Apart from the future design exercise and a prompt to consider the viewpoints of the people living in the year 2050 during the deliberation, the treatment condition followed the same deliberative process as the control condition. An initial overview of the the data regarding group discussions suggests that participants in the future design treatment reacted to the exercise in varying ways. At least some participants referred to the future design exercise or reported having thought about the exercise during the discussions.

The collection of the data used in this study began one week before the assembly, when those who had volunteered to participate in it filled in a pre-test survey. In this survey, the participants were asked to prioritize goals related to the future of the region. In addition, this survey included questions on taking future generations' perspectives and willingness to make sacrifices for future generations, which are analysed in this article, as well as other questions measuring participants' attitudes towards long-term decision-making. After the pre-test survey, participants received a document containing background information about the current state of the region, including information on demographic, economic, environmental and other matters. The post-test survey included the same questions as the pre-test survey and questions about the respondents' evaluation of quality of discussions. The post-test survey was divided into two parts; the first part was answered during the break and the second part right after the event. All participants completed the pre-test survey ($n = 55$) while one failed to complete the first part of the post-test survey ($n = 54$) and two failed to complete the second part ($n = 53$).

## 4. Results

The first part of the empirical analysis takes a closer look at how participation in the Citizens' Assembly affects the aptitude to take future generations' perspectives. We want to know to whether democratic deliberation increases proneness to perspective-taking, and whether the time travel exercise enhances such proneness further (Hypotheses 1 and 3). To this end, we make use of pre/post-test design with attitude measurements before and after the assembly. To examine the effects of participation in the Citizens' Assembly and the differences between future design treatment and regular deliberation treatment, we compare means of various measures in the pre-test and post-test surveys. Paired samples t-tests are used to inspect changes across the whole sample and within treatments, and independent samples t-tests are used to compare treatment and control.

To measure perspective taking aptitude, we use the following three items: "I often think about what kind of world we are leaving to future generations (Translated from original in Finnish: "Mietin paljon sitä, minkälaisen maailman jätämme tuleville sukupolville.")", "When I read or see a story set in the future, I imagine the events hap-

pening to me (Translated from original in Finnish: "Kun luen tai katselen tulevaisuuteen sijoittuvaa tarinaa, kuvittelen miltä tuntuisi, jos kokisin tarinan tapahtumat itse.")", and "I sometimes try to envision the future by imagining what things look like from the perspective of future generations (Translated from original in Finnish: "Yritän toisinaan hahmottaa tulevaisuutta paremmin kuvittelemalla, miltä asiat näyttäisivät jälkipolviemme näkökulmasta.")". These items represent different ways the respondents think about future generations in their daily lives. In addition, we make use an index combining the three items. The findings for these measures are reported for each of the deliberative treatments and for all participants in the deliberative experiment. Following Hypotheses 1 and 3 we expect that the participants' perspective-taking aptitude would increase as a result of taking part in deliberation, and that the time travel exercise in combination with deliberation (future design treatment) would further increase this aptitude.

The findings from paired samples T-tests for the perspective taking aptitude (Table 2) are mixed at best. On the one side there is clearly significant change in the mean for the third item measuring perspective taking, both for all participants (at the 0.05 level) and for the future design treatment (at the 0.01 level). A closer examination reveals that it is the future design treatment that is driving the results, and there is no evidence of similar change taking place in the regular deliberation treatment. The perspective-taking index also shows a significant mean change between the two measuring points, but again this outcome is mostly a result of the notable change that takes place in the third item among participants in the future treatment. It should also be noted that the third item is measuring behavior very similar to the time travel exercise assigned to the future treatment participants. In this respect, this analysis gives only weak support that deliberation in the Satakunta2050 Citizens' Assembly increased the perspective-taking aptitude among the participants.

**Table 2.** Perspective-taking capacity of participants in Satakunta2050.

| | All Participants (N = 54) | | Future Design Treatment (N = 28) | | Regular Deliberation Treatment (N = 26) | |
|---|---|---|---|---|---|---|
| | Mean t1 | Change t1–t2 | Mean t1 | Change t1–t2 | Mean t1 | Change t1–t2 |
| i. "I often think about what kind of world we are leaving to future generations" | 0.56 | 0.01 | 0.56 | 0.03 | 0.57 | −0.01 |
| ii. "When I read/see story set in future, I imagine the events happening to me" | 0.49 | −0.02 | 0.52 | 0.01 | 0.45 | −0.06 |
| iii. "I sometimes try to envision the future, by imagining what things look like from the perspective of future generations" | 0.48 | **0.05 *** | 0.44 | **0.10 **** | 0.52 | 0.00 |
| Perspective taking-index (a + b + c) | 1.53 | 0.03 | 1.52 | **0.13 *** | 1.53 | −0.07 |

Significance: † <0.10, * <0.05, ** <0.01. Variables for statement a and b have been coded to vary between 0–1. Variables a, b and c are highly correlated (0.34–0.60, $p < 0.001$) and have also been combined into a perspective taking-index.

To further test Hypotheses 1 and 3, additional between-group comparisons were made utilizing two variables from the post-test survey. Using an 11-point scale ranging from 'very hard' to 'very easy', participants answered the questions 'How easy was it for you to imagine how life would be like in Satakunta in 2050?' and 'How easy was it for you to take the future generations' perspective?'. Assuming Hypothesis 3 would hold true, means for the abovementioned questions should be higher in the future design treatment compared to regular deliberation. Results from two-sided between-group T-tests indicate that this is not the case, however. Table 3. shows that, first, the mean difference between the treatments is not significant with either of the variables, and second, that the means of

future design treatment are actually lower than those of the regular deliberation. While the high *p*-values suggest that this might be due to pure coincidence, it could be possible that the people in future design groups found imagining the future and taking the perspective of future generations harder precisely because they had tried it themselves. One way or the other, Hypothesis 3 is not supported by the data from our post-test survey.

**Table 3.** Perspective-taking and imagining the future.

| | Future Design Treatment (N = 27) | Regular Deliberation Treatment (N = 26) | |
|---|---|---|---|
| | Mean | Mean | Mean difference (sig.) |
| How easy was it for you to imagine how life would be like in Satakunta in 2050? (0–10) | 5.63 | 6.12 | 0.486 (0.412) |
| How easy was it for you to take the future generations' perspective? (0–10) | 5.52 | 6.08 | 0.558 (0.362) |

The Hypotheses 1 and 3 can be further tested through examining participants' responses to a question measuring their willingness to consider future generations' perspectives: 'From whose perspective should the future of Satakunta region be planned primarily?' The answers were given on an 11-point scale, where 0 meant that the future of Satakunta should primarily be planned from the perspective of *current* inhabitants, and 10 meant that it should be planned from the perspective of *future* inhabitants. To explore the effects of deliberation, the mean difference of pre-test and post-test answers was analyzed using two-tailed paired samples T-test.

As can be seen in Table 4, deliberation appears to have had a noteworthy effect on participants' answers. The mean position of all participants increased by over 0.8 points from 5.74 to 6.55, and the change is significant at a 0.01 level. This effect can also be observed in both treatment groups separately, although with slight differences in volume and significance levels. The results indicate that deliberation, especially when combined with the time travel exercise, indeed supported the willingness to consider future generations' perspectives in regional planning and thus provide support for Hypothesis 1. However, independent samples T-test (not reported here) shows no significant difference in mean change between the treatment groups, which does not provide support for the Hypothesis 3 pertaining to the effects of the future design treatment.

**Table 4.** From whose perspective should the future of Satakunta region be planned primarily?

| | All Participants (N = 53) | | Future Design Treatment (N = 27) | | Regular Deliberation Treatment (N = 26) | |
|---|---|---|---|---|---|---|
| | Mean t1 | Change t1–t2 | Mean t1 | Change t1–t2 | Mean t1 | Change t1–t2 |
| From whose perspective should the future of Satakunta region be planned primarily? | 5.74 | **0.811 \*\*** | 5.41 | **0.852 \*** | 6.08 | **0.769 [†]** |

Significance: [†] <0.10, \* <0.05, \*\* <0.01. 0 = from the perspective of people currently living in Satakunta—10 = from the perspective of people who will live in Satakunta in the future.

We now turn to examine the Hypotheses 2 and 4 pertaining to the effects of deliberation and future design on willingness to make sacrifices for future generations, which is considered as the key component of sustainability [1]. In other words, we are interested in participants' readiness to accept costs in case of intergenerational conflicts. Willingness to make sacrifices for future generations is measured with the help of two items 'Today's voters should be prepared to reduce their standard of living for the wellbeing of future

generations (Translated from original in Finnish: "Nykyäänestäjien tulisi olla valmiita tinkimään elintasostaan, jos tulevien sukupolvien hyvinvointi sitä vaatii.")' and 'I would be willing to pay more taxes if we thereby can improve the well-being of future generations (Translated from original in Finnish: "Voisin maksaa korkeampia veroja, mikäli siten voidaan parantaa tulevien sukupolvien hyvinvointia.")', as well as an index combining the two items. The first item is about making sacrifices as a collective by lowering standards of wellbeing, while the second is about personal willingness to contribute to collective efforts by paying taxes. Following Hypothesis 2 we expect that deliberation in general would lead to a greater willingness to make sacrifices for future generations, and based on Hypothesis 4 we expect that taking part in the future design exercise before deliberating would further increase the willingness to make sacrifices for future generations.

From the results reported in Table 5, we can see that there is some evidence to support the former hypothesis, and that the results overall are more consistent than for the perspective-taking aptitude. There appears to be a modest increase in the willingness to make sacrifices across the board but because the changes are relatively small, the findings are statistically significant only when adding all participants together. When looking at all participants the mean change for the first item 'Today's voters should accept . . . ' is significant at the 0.10 level and the combined measure is significant at the 0.05 level. There are no significant changes for neither the future design treatment nor the regular deliberation treated separately.

**Table 5.** Willingness to make sacrifices for future generations among participants in Satakunta2050.

| | All Participants (N = 54) | | Future Design Treatment (N = 28) | | Regular Deliberation Treatment (N = 26) | |
|---|---|---|---|---|---|---|
| | Mean t1 | Change t1–t2 | Mean t1 | Change t1–t2 | Mean t1 | Change t1–t2 |
| i. Today's voters should be prepared to reduce their standard of living for the wellbeing of future generations. | 0.46 | **0.04** [†] | 0.45 | 0.03 | 0.48 | 0.03 |
| ii. I would be willing to pay more taxes if we thereby can improve the well-being of future generations. | 0.29 | 0.02 | 0.26 | 0.00 | 0.31 | 0.06 |
| Make sacrifices-index (a + b) | 0.76 | **0.06** * | 0.72 | 0.04 | 0.80 | 0.08 |

Significance: [†] <0.10, * <0.05, ** <0.01. Variables for statement a and b have been coded to vary between 0–1. Variables a and b are highly correlated (0.41, $p < 0.001$) and have also been combined (a + b) into a sacrifice-index.

We also inspect willingness to make sacrifices for future generations by analyzing support for a concrete policy measure that involves an intergenerational conflict. According to our Hypotheses 2 and 4, deliberation and especially deliberation enhanced with future design exercise should increase the support for future-oriented policy. Previous research has found that when presented with different concrete policies, people can be supportive even for options that bring greater benefits to the generations that come after them [36]. We asked participants about their preferences on a concrete policy to protect against destructive floods that are a recurring natural hazard in the Satakunta region. The region's largest river, Kokemäenjoki, and its estuary belong to the areas most prone to flooding in Finland.

Participants were asked to select one of three options, each with a different distribution of benefits between their generation and the next generation. Option A prevents 20 destructive floods during this generation, and 10 during the next generation, Option B prevents no destructive floods during this generation, but 30 during the next generation, Option C prevents 10 destructive floods during this generation, and 20 during the next generation.

Note that we did not make available an 'equal shares' option as it has been found to introduce bias. When allowed to allocate resources equally between the present and future generations, most respondents may opt for this option simply in order to avoid the moral weighting between the importance of well-being of future others and present citizens [36].

Options that clearly favor either the present or the future generation could thus be better in measuring true presentist or selfish attitudes that are not as socially desirable as sharing costs or burdens equally. This variable was recoded into a new measure of support for policy choices, where the most presentist option was given value 0, modestly future-regarding option value 0.5 and the most altruistic option from the perspective of future generations value 1.

As we see in Table 6, there was a minor increase in the future design treatment in support for preventing more destructive floods during the next generation than during this generation, but the changes were not statistically significant even at 0.10 level. Furthermore, in the regular deliberative process treatment support for long-term policy actually decreased, which means that there was no significant change even when looking at the full sample. Hypotheses 2 and 4 are therefore not supported by our data on policy preferences in an issue framed as an intergenerational conflict in flood protection.

**Table 6.** Policy preferences for flood protection among participants in Satakunta2050.

| | All Participants (N = 54) | | Future Design Treatment (N = 28) | | Regular Deliberation Treatment (N = 26) | |
|---|---|---|---|---|---|---|
| | Mean t1 | Change t1–t2 | Mean t1 | Change t1–t2 | Mean t1 | Change t1–t2 |
| Support for future-regarding policy | 0.42 | −0.02 | 0.41 | 0.02 | 0.42 | −0.06 |

Significance: [†] <0.10, * <0.05, ** <0.01. Variable has been coded to vary between 0–1.

## 5. Discussion

There is a widely recognized need to develop institutions and practices that can prompt individuals to think in more intergenerational terms, in order to effectively implement the required sustainability transformations in democratic societies. Deliberative mini-publics are expected to have a capacity to enhance, not just participants' understanding of long-term consequences of policy choices, but also their capacity to perspective-taking and sense of intergenerational fairness. Obviously, mini-publics are just one possible remedy to the problem of short-termism in representative politics. Moreover, there are open questions regarding the possible roles of mini-publics or other randomly selected bodies in representative democracies [6,24].

In this article, we have examined the effects of participation in a deliberative mini-public on citizens' willingness and capacity to take future generations' perspectives and care for their well-being The effects were rather clear when it comes to participants' overall willingness to consider future generations' perspectives. Deliberative process made participants increasingly think that the regional planning process should be conducted from the perspectives of people living in the year 2050. This result seems to confirm the view that deliberation can encourage people to accept and support sustainability transformations in public decision-making, as expected. At the same time, our findings suggest that it is hard for individuals to imagine future generations' perspectives [37], even when particularly encouraged to do so. More specifically, our results suggest that while deliberation as such did not improve participants' capacity to step in future generations' shoes, the mental time travel treatment had only a modest positive effect in this respect. Yet, considering the level of abstraction when discussing the perspectives of people living in the year 2050, perhaps it is not a wonder that we did not observe stronger effects in terms of aptitude of taking future generations' perspectives.

In terms of willingness to make sacrifices on behalf of future generations, deliberation seems to have had a small, positive effect. Most notably, the different deliberative treatments did make participants more prone to accept that there may be a need for current generations' to make sacrifices in order to ensure future generations' well-being. In other words, there seems to be a potential in deliberation to make citizens accept some burdens, including tax increases, for the benefit future generations. This finding suggests that de-

liberation can help evoke sense of intergenerational fairness and make more sustainable policy choices, which is in line with findings by MacKenzie and Caluwaerts [24]. However, the results regarding the flood protection question suggest that democratic deliberation does not necessarily increase present generations' willingness to make altruistic sacrifices in intergenerational conflicts.

The results of our experiment seem to be partly in line with but partly different from the observations of Hara et al. [12]. There could be several reasons for these differences. It could be that, in order to achieve similar results, more emphasis needs to be put on the content and breadth of the future design itself. While the participants in Hara et al. study were tasked to actually *represent* these interests throughout the deliberative process, in our study the participants were asked, first, to imagine themselves in the future and, second, to consider future perspectives during the deliberative process. In addition, the deliberative process Hara et al. [12] was longer and was conducted face-to-face, while ours was a one-day process conducted online. That said, Hara et al. [12] relied on a very limited sample of participants and, unlike our study, it did not make use of an experimental design that provides a more reliable test of the mechanisms driving future-oriented behavior.

Nevertheless, it must be admitted that the relatively small number of participants in the Citizens' Assembly calls for some caution when interpreting our results, especially in terms of external validity. Further research is thus needed to explore how different design features of deliberative mini-publics—e.g., duration, information, and inclusion of youth perspectives—might facilitate consideration of future generations' perspectives and interests in the context of deliberative mini-publics. Further studies should also investigate the capacity of mental time travel exercises and other similar interventions to change participants' mindsets concerning future generations.

Our experimental study gives some support to the view that deliberation on policy goals as well as hearing the perspectives of people with different backgrounds can help participants understand the trade-offs related to long-term policy-making, or at least become aware of them. Deliberation in the Citizens' Assembly made participants more committed to consider future generations' perspectives and interests in long-term planning and somewhat more likely to accept costs to ensure future generations' well-being. However, the task of imagining the particular perspectives of future people, in itself, did not seem to become any easier in the process.

**Author Contributions:** All authors have contributed to the planning and the implementation of the experiment and the survey, the analysis of the data, and writing up the article. All authors have read and agreed to the published version of the manuscript.

**Funding:** This research has been funded by Strategic Research Council project "Participation in Long-Term Decision-Making", decision number: 312671.

**Institutional Review Board Statement:** The study was conducted according to the guidelines of the Declaration of Helsinki.

**Informed Consent Statement:** Informed consent was obtained from all subjects involved in the study.

**Data Availability Statement:** The data will be made available at the Finnish Social Science Data Archive (FSD) after an embargo.

**Conflicts of Interest:** The authors declare no conflict of interest.

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
