# Peer review of "For the Sake of the Future: Can Democratic Deliberation Help Thinking and Caring about Future Generations?"

_sustainability, doi:10.3390/su13105487_

Round 1
Reviewer 1 Report
This is a good article with an interesting and novel premise. The argument is internally coherent, and I think the arguments put forward are generally supported by the evidence shown.
However, I think the article introduction, discussion and conclusion need to better situate the findings of this study, to make clear what their implications are, and why the authors felt it was a worthwhile study to carry out.
The work of people like Alexander Bogner points out that deliberative workshops are very controlled and unusual environments. So I think there are a few extra logical steps which the authors need to make to translate the findings into a more general argument.
Is the proposal that deliberative engagement should be deliberately embedded in a range of policy and decision-making contexts in order to ensure that future generations are accounted for? If so then how do we know that this is a better method for accounting for the needs of future generations than methods such as youth engagement (and, in particular listening to the vast numbers of young people who are already making their voices heard on this issue), formal foresight processes, and cost-benefit analysis?
Or would the authors instead posit that the study shows that most people have the capacities to more meaningfully account for the needs of future generations in the right context and with the right prompts? In this case, are there any more general lessons to be learned about how these concerns can be better integrated into representative democracies (which the authors rightly point out have short-termist tendencies)?
In order to bring out these broader implications of the project findings it might be useful to draw on some of the more qualitative data produced through the assembly. For example, how did the participants talk about and frame the concerns of future generations in their deliberations? Were there any common assumptions or themes which came up? Are these similar or different to the ways in which other methods of appraisal try to account for futures?
Author Response
This is a good article with an interesting and novel premise. The argument is internally coherent, and I think the arguments put forward are generally supported by the evidence shown.
However, I think the article introduction, discussion and conclusion need to better situate the findings of this study, to make clear what their implications are, and why the authors felt it was a worthwhile study to carry out.
The work of people like Alexander Bogner points out that deliberative workshops are very controlled and unusual environments. So I think there are a few extra logical steps which the authors need to make to translate the findings into a more general argument.
Is the proposal that deliberative engagement should be deliberately embedded in a range of policy and decision-making contexts in order to ensure that future generations are accounted for? If so then how do we know that this is a better method for accounting for the needs of future generations than methods such as youth engagement (and, in particular listening to the vast numbers of young people who are already making their voices heard on this issue), formal foresight processes, and cost-benefit analysis?
We would like to thank the reviewer for relevant comments, which we have taken into account when revising the paper. The reviewer is correct in pointing out that we did not discuss the broader implications of our findings very much. We have now revised the Discussion so that we contextualize our study better. More specifically, we describe the need to identify and develop institutions that prompt individuals to think in more intergenerational terms. We also point out that our results highlight the need for further research on how deliberative mini-publics in general could be part of this repertoire of institutions.
Or would the authors instead posit that the study shows that most people have the capacities to more meaningfully account for the needs of future generations in the right context and with the right prompts? In this case, are there any more general lessons to be learned about how these concerns can be better integrated into representative democracies (which the authors rightly point out have short-termist tendencies)?
This is also a valid point, and we have now mentioned the need to explore the role of deliberative mini-publics and other future-oriented institutions in the context of representative democracies.
In order to bring out these broader implications of the project findings it might be useful to draw on some of the more qualitative data produced through the assembly. For example, how did the participants talk about and frame the concerns of future generations in their deliberations? Were there any common assumptions or themes which came up? Are these similar or different to the ways in which other methods of appraisal try to account for futures?
We agree with the reviewer that qualitative data can be informative for delving more deeply into the process of deliberation in a citizen assembly, but an adequate content analytical or qualitative analysis of the discussions is not within the scope of this article. However, we have added a sentence regarding the effects of the “future design” exercise on small group deliberations (lines 294-297).
Reviewer 2 Report
Report on "For the Sake of the Future: Can democratic deliberation help thinking and caring about future generations?"
This article reports the results of a deliberative experiment conducted as part of the deliberations of a Citizens' Assembly in the Satakunta region of Finland. Four hypotheses about the capacity of deliberation to stimulate concern for future generations and incentivize sacrifices in the present for future benefits were tested, though only two of these, involving "mental time travel", are strictly experimental manipulations (the other two did not have a control group). The study was nicely designed to take advantage of the deliberations of the Citizen's assembly and its results are clearly presented, though I am concerned that they are also overstated; even if they did not occur by chance, the effects of deliberation on taking the perspective of future generations and being willing to make sacrifices for them are so small that they seem unlikely to have any external validity.
In particular, the abstract argues that "Our findings suggest that deliberation as such increases participants’ willingness to consider future generations’ perspectives in long-term planning, and consideration of those perspectives can be boosted somewhat by the mental time travel exercise". But of the results presented, deliberation had a statitically significant effect for only two items of eight survey items tested; given the statistical problems of multiple comparisons, it is hard to take this result as anything but noise, especially given the small sample size (55 people). Even if we assume the effects are real, the magnitude is small (five percentage points in a survey item on whether the subjects "try to envision the future, by imagining what things look like from
the perspective of future generations", which could easily be due to *one* subject changing their answer between pre- and post- survey, if I understand the design correctly). Moreover, the experimental manipulation (time travel) did not seem to have any distinctive impact except in one item (the one that asked people whether they try to envision the future, which as the authors note may be explained by the fact that participants in the time travel exercise where in fact asked to imagine the future), and it is not clear that its effect in the "perspective" question (table 4) is due to the manipulation itself or the general deliberation (no "difference of means" test is reported between the control and treatment groups). So I do not think the study can claim that "consideration of those perspectives can be boosted somewhat by the mental time travel exercise".
This doesn't mean that deliberation didn't have an effect, only that the effect as treported seems a bit overstated, and insufficiently supported by the results reported. Beyond these considerations I find it difficult to believe that the very small attitudinal changes reported in table 5 would be sustained beyond the setting of the assembly, but that is a problem of external validity that the authors do not need to tackle (though perhaps they could note).
The writing is generally clear and free of grammatical or stylistic issues, but I will note the following:
Line 23: "The need and urgency of sustainability transformations presents democratic societies an unforeseen challenge" should be "The need and urgency of sustainability transformations presents democratic societies with an unforeseen challenge"
Line 42 "so-called deliberative mini-publics or in randomly selected second chambers, seems to have" should be "seem to have"
Line 69 "to help taking future generations’ perspectives and consideration of their interests" should be "to help take ... and consider"
Line 397 "a recurring nature hazard in the Satakunta region" should be "a recurring natural hazard in the Satakunta region"
Caption of table on p. 11 "Variables and b are highly correlated" should be "Variables a and b are highly correlated"
Author Response
This article reports the results of a deliberative experiment conducted as part of the deliberations of a Citizens' Assembly in the Satakunta region of Finland. Four hypotheses about the capacity of deliberation to stimulate concern for future generations and incentivize sacrifices in the present for future benefits were tested, though only two of these, involving "mental time travel", are strictly experimental manipulations (the other two did not have a control group). The study was nicely designed to take advantage of the deliberations of the Citizen's assembly and its results are clearly presented, though I am concerned that they are also overstated; even if they did not occur by chance, the effects of deliberation on taking the perspective of future generations and being willing to make sacrifices for them are so small that they seem unlikely to have any external validity.
In particular, the abstract argues that "Our findings suggest that deliberation as such increases participants’ willingness to consider future generations’ perspectives in long-term planning, and consideration of those perspectives can be boosted somewhat by the mental time travel exercise". But of the results presented, deliberation had a statitically significant effect for only two items of eight survey items tested; given the statistical problems of multiple comparisons, it is hard to take this result as anything but noise, especially given the small sample size (55 people). Even if we assume the effects are real, the magnitude is small (five percentage points in a survey item on whether the subjects "try to envision the future, by imagining what things look like from
the perspective of future generations", which could easily be due to *one* subject changing their answer between pre- and post- survey, if I understand the design correctly). Moreover, the experimental manipulation (time travel) did not seem to have any distinctive impact except in one item (the one that asked people whether they try to envision the future, which as the authors note may be explained by the fact that participants in the time travel exercise where in fact asked to imagine the future), and it is not clear that its effect in the "perspective" question (table 4) is due to the manipulation itself or the general deliberation (no "difference of means" test is reported between the control and treatment groups). So I do not think the study can claim that "consideration of those perspectives can be boosted somewhat by the mental time travel exercise".
We would like to thank the reviewer for these important points. The reviewer is correct in that the abstract was overselling our findings. We have now revised the abstract and updated our conclusions to better reflect the results of our analyses. In tables 2 and 4, we report only the pre-test values and the changes because the pre-test values were somewhat different in treatments (by chance).
This doesn't mean that deliberation didn't have an effect, only that the effect as treported seems a bit overstated, and insufficiently supported by the results reported. Beyond these considerations I find it difficult to believe that the very small attitudinal changes reported in table 5 would be sustained beyond the setting of the assembly, but that is a problem of external validity that the authors do not need to tackle (though perhaps they could note).
We have revised the section 'Discussion' so that we now clearly acknowledge that our results give only modest support to our hypotheses. Consequently, we have also highlighted the need for further research. The issue of external validity is also mentioned in the Discussion.
The writing is generally clear and free of grammatical or stylistic issues, but I will note the following:
Line 23: "The need and urgency of sustainability transformations presents democratic societies an unforeseen challenge" should be "The need and urgency of sustainability transformations presents democratic societies with an unforeseen challenge"
Line 42 "so-called deliberative mini-publics or in randomly selected second chambers, seems to have" should be "seem to have"
Line 69 "to help taking future generations’ perspectives and consideration of their interests" should be "to help take ... and consider"
Line 397 "a recurring nature hazard in the Satakunta region" should be "a recurring natural hazard in the Satakunta region"
Caption of table on p. 11 "Variables and b are highly correlated" should be "Variables a and b are highly correlated"
Thank you for pointing out these grammatical and stylistic errors. They have now been corrected accordingly. In addition, we have made grammar and stylistic corrections throughout the text.